# Towards Online 3D Bin Packing: Learning Synergies between Packing and Unpacking via DRL

**Shuai Song, Shuo Yang, Ran Song, Shilei Chu, Yibin Li, Wei Zhang**[*]

Shandong University

{shuaisongss, shuoyang, slchu}@mail.sdu.edu.cn, {ransong, liyb, davidzhang}@sdu.edu.cn

**Abstract:** There is an emerging research interest in addressing the online 3D bin packing problem (3D-BPP), which has a wide range of applications in logistics industry. However, neither heuristic methods nor those based on deep reinforcement learning (DRL) outperform human packers in real logistics scenarios. One important reason is that humans can make corrections after performing inappropriate packing actions by unpacking incorrectly packed items. Inspired by such an unpacking mechanism, we present a DRL-based packing-and-unpacking network (PUN) to learn the synergies between the two actions for the online 3D-BPP. Experimental results demonstrate that PUN achieves the state-of-the-art performance and the supplementary video shows that the system based on PUN can reliably complete the online 3D bin packing task in the real world.

**Keywords:** Bin packing, robotics, deep reinforcement learning.

## 1   Introduction

The rapid growth of e-commerce has significantly increased the burden of human packers in logistics warehouses. Thus there is an emerging research interest in developing intelligent robotic systems to replace human labors in the repetitive work [1]. In a typical packing scenario, the human packer is asked to pick the items from a running conveyor belt one by one and then pack them into a bin with as higher space utilization as possible. To solve this task, most researchers formulated it as the online 3D bin packing problem (3D-BPP) and presented algorithms to optimize the bin packing policy [2, 3, 4]. However, as one of the most classic combinatorial optimization problems, the online 3D-BPP is notoriously NP-hard [5]. It is nontrivial to find the optimal policy as only one item is observable each time.

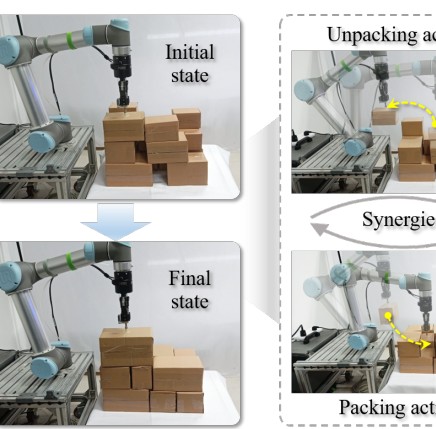

Figure 1: Illustration of the synergies between packing and unpacking actions. Our method can unpack the inappropriately placed items and repack them into the bin to improve the overall space utilization.

Early methods for the bin packing problem mostly focus on designing reasonable bin packing heuristics [6, 7, 8]. Such heuristics are essentially hand-designed rules abstracted from the experiences of human packers. Although this is an intuitive way to solve the bin packing task and usually maintains good working efficiency, the experiences of human packers are not always perfect. Moreover, without a learning process, the heuristic algorithms are less likely to find the optimal bin packing policy in various scenarios.

Inspired by the recent advances in deep reinforcement learning (DRL) [9, 10, 11], researchers attempt to solve the online 3D-BPP via DRL [12, 13, 14]. Although these methods have achieved promising results in the bin packing task, they neglected the crucial human experience of unpacking when the currently observed item cannot be packed into an appropriate position. The ability of un-

---

[*]Corresponding author

6th Conference on Robot Learning (CoRL 2022), Auckland, New Zealand.

packing is thus particularly important for the online 3D-BPP, where the agent can only observe one upcoming item and hence is less likely to learn the optimal bin packing policy.

Therefore, we introduce the unpacking mechanism for the online 3D-BPP as shown in Fig. 1. To learn the synergies between packing and unpacking, we propose a packing-and-unpacking network (PUN) based on DRL. Specifically, PUN is a two-branch architecture which generates the packing and the unpacking actions and their corresponding state values. It then determines to perform either the packing or the unpacking action based on the state values. Finally, the synergies between packing and unpacking actions are learned via DRL. We also design packing and unpacking heuristics and incorporate them into the DRL framework.

The main contributions of this paper are twofold: 1) we introduce the unpacking mechanism for online 3D-BPP and propose to solve it with both the packing and unpacking actions; 2) we present PUN to learn the synergies between packing and unpacking actions via DRL in the simulated environment, seeking for the optimal bin packing policy. And a bin packing system developed based on PUN is demonstrated in the real world.

## 2   Related Work

Existing methods for the 3D-BPP can be divided into two groups: heuristic and DRL-based methods. We herein briefly review both of them.

**Heuristic methods.** The 3D bin packing problem (3D-BPP) focuses on packing a set of cubical items into a confined 3D space (e.g. a shipping box) with a high rate of space utilization [15]. It is a classic NP-hard optimization problem [5]. The online 3D-BPP assumes that only the current item to be packed is observable while the upcoming items are unknown. Early work mostly designed different heuristic algorithms [16, 17] such as Tabu Search [18], First-Fit [19] and Extreme Point-Based Heuristics [7]. The heuristic algorithms are essentially the distillation of the real-world packing experience of human packers, which are not always perfect for various scenarios.

**DRL-based methods.** Recently, DRL has been demonstrated effective to solve combinatorial optimization problems [20, 21, 22, 23]. Some researchers thus attempted to solve the 3D-BPP using DRL [12, 13, 24, 14, 25]. For example, Zhao et al. [13] formulated the 3D-BPP as constrained Markov decision process and first employed the DRL method to optimize the sequence of the items to be packed into the bin. Verma et al. [24] developed a DRL algorithm to solve the online 3D-BPP for an arbitrary number of bins with various sizes. More recently, Yang et al. [14] made the primary attempts to incorporate the heuristics into the DRL framework for learning more optimal bin packing policy. However, such methods cannot perform comparably well with the human packers.

A neglected human experience is that human packers can unpack some item from the bin if packing the current item is difficult. We thus incorporate the unpacking mechanism into the DRL framework which learns the synergies between the packing and the unpacking actions.

## 3   Method

In an online bin packing task, the items are delivered from a running conveyor belt one by one and then packed into a bin. Only the immediately incoming item is observable. We formulate the bin packing task as a Markov Decision Process (MDP) [26] subject to the state $\mathcal{S}$, the action $\mathcal{A}$, the transition $\mathcal{P}$, and the reward $\mathcal{R}$. We solve the MDP with an end-to-end DRL framework which seeks for a policy $\pi(a_t|s_t; \theta_\pi)$ to maximize the sum of the expected rewards, expressed as:

$$J_{\pi_\theta} = \max_\theta \mathbb{E}_{s,a \sim \pi_\theta} \left[ \sum_{t=1}^{T} \gamma^{t-1} r\left(s_t, a_t\right) \right].$$ (1)

This section provides details of PUN and the implementation of a real-world logistics application.

### 3.1   Packing-and-Unpacking Network

The architecture of the proposed PUN is illustrated in Fig. 2 and elaborated as follows.

**State representation.** We model the state representation $s_t$ as the current configuration of the bin and the current item to be packed. To parameterize the configuration of an $L \times W \times H$ bin,

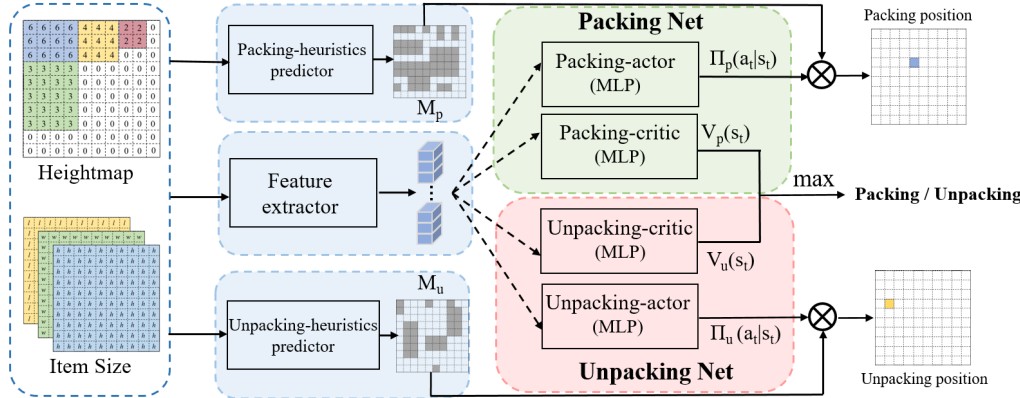

Figure 2: Illustration of the proposed PUN for learning synergies between packing and unpacking.

we discrete the state observation of the bin at the time $t$ as a heightmap $\mathbf{H}_t \in \mathbb{Z}^{+L \times W}$, where $\mathbb{Z}^+ \in [0, H]$. The heightmap is an $L \times W$ discrete grid and the value at each cell represents the height of the stacked items at that position. The state of the current item $n$ with the size of $l_n \times w_n \times h_n$ is represented as an $L \times W \times 3$ map, the 3 channels of which are assigned with the values of $l_n, w_n$ and $h_n$ respectively (see Fig. 2).

**Action definition.** We parameterize each action $a_t$ as a motion primitive behavior $\psi$ (e.g. packing or unpacking) executed at the position $p$ projected from a cell $c$ of the heightmap representation of the bin configuration:

$$a_t = (\psi, p) \mid \psi \in \{pack, unpack\}, p \to c \in \mathbf{H}_t. \tag{2}$$

Each cell $c$ in the heightmap of the bin configuration corresponds to a specific position to execute the packing or unpacking action. We use the left-front-bottom corner of an item as its packing or unpacking coordinates. The motion primitive behaviors are defined as below:

- Packing: $p$ denotes putting the item at the cell $c$ of the grid corresponding to the bin heightmap. We use a 2D coordinates $(x_n, y_n)$ to represent the position of $c$. The action $(pack, p_{(x_n, y_n)})$ denotes that the agent puts the item $n$ at the position $(x_n, y_n)$ of the heightmap.

- Unpacking: $p$ denotes removing the item from the cell $c$ of the grid corresponding to the bin heightmap. We set a temporary item buffer to store at most $B$ unpacked items. The action $(unpack, p_{(x_m, y_m)})$ denotes that the agent removes the item $m$ from the position $(x_m, y_m)$ of the heightmap and places it in the temporary item buffer.

**Synergies between packing and unpacking actions.** As illustrated in Fig. 2, PUN is designed as a two-branch network to learn the synergies between packing and unpacking. It takes as input the heightmap of the bin configuration and the sizes of the item, and outputs the motion primitive behavior $\psi$ (packing or unpacking) executed at the position $p$ projected from a cell of the heightmap. The state input is encoded into features by a shared feature extraction module which is implemented with 5 fully-connected layers. Such features are then fed into to the packing and the unpacking networks, respectively.

The packing and unpacking branches both consist of three major components: an actor network, a critic network and a heuristic mask predictor. The actor network outputs the probability distribution of the motion primitive behavior (packing and unpacking) with the same resolution as that of the state $s_t$. The two critic networks predict the state values $V_p(s_t)$ and $V_u(s_t)$ which measure the cumulative reward of taking the two motion primitive behaviors in the state $s_t$ at the time $t$ respectively [27]. The heuristic mask predictors generate the packing and the unpacking masks $M_p$ and $M_u$ to estimate valid packing and unpacking positions based on the heuristic rules. The probability distribution output by the actor network is modulated by the heuristic mask.

For the state $s_t$ at the time step $t$, the network outputs two state values $(V_p(s_t), V_u(s_t))$ and two probability distributions $(\pi_p(a_t|s_t), \pi_u(a_t|s_t))$ corresponding to packing and unpacking actions.

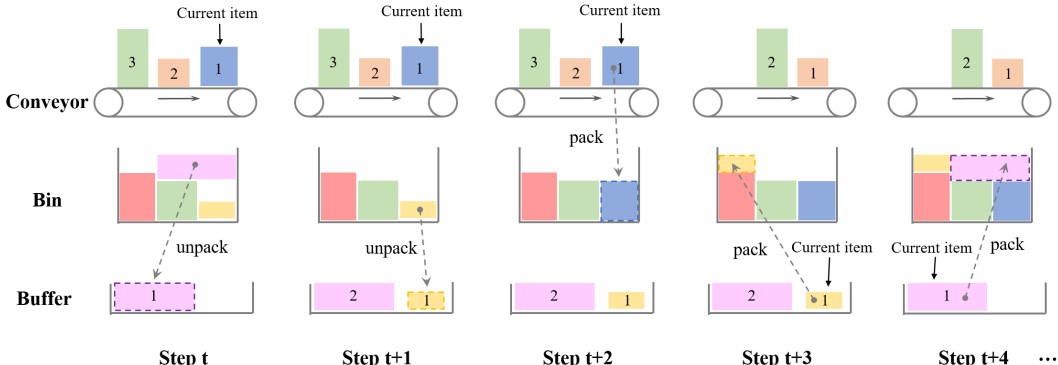

Figure 3: An example of the synergies between packing and unpacking. Current item indicates the item to be packed at the current time step. The serial number on the item indicates the order in which it is to be packed.

One action is subject to the motion primitive behavior $\psi$ and the position $p$. The final motion primitive behavior is the one with the highest state value between the two primitive behaviors, which means that the DRL agent will theoretically earn the highest rewards. The position is sampled from the probability distribution corresponding to the final motion primitive behavior.

The synergistic process of packing and unpacking is elaborated as follows. For the item $n$ to be packed at the time step $t$, PUN takes its size and the heightmap of the bin as input to predict the motion primitive behavior $\psi$ and the target position $p$ of the bin heightmap. If the packing action is predicted, the agent will pick the item $n$ from the conveyor belt and place it at the target position $p$, and take the next item from the conveyor belt as the item to be packed at the time step $t + 1$. If the unpacking action is predicted, the agent will pick the item $m$ from the target position $p$ and place it into the buffer zone to temporarily store it. The agent will continue to predict the packing or unpacking action for the item $n$ at the next time step until packing it into the bin. After packing the item $n$, the item $m$ in the temporary buffer will be used as the current item to be packed for predicting the next action. If there are multiple items in the temporary buffer, the agent will select the target item to be packed according to the order in which the items were unpacked into the temporary buffer (first in last out). Only after all items in the temporary buffer are emptied can the robot select the next item to be packed from the conveyor belt. Fig. 3 shows an example of the process of packing and unpacking.

**Heuristics-based action constraint.** We incorporate three heuristic rules into the DRL framework to guide the reinforcement learning. The first one is physics heuristics. It defines the actions that lead to failure as invalid actions. For example, the packing actions that pack the items beyond the bin boundaries and the unpacking actions that execute on the positions without items. The second one is packing heuristics. It consists of four popular packing heuristic algorithms which are described as follows: The **Extreme Point algorithm** [7] that calculates all extreme points in the bin and packs item based on extreme point rule. The **Empty Maximal Space algorithm** [8] that packs item in the position which can produce the largest empty orthogonal spaces in the bin. The **First Fit algorithm** [19] that packs item in the first feasible position. The **Floor Building algorithm** [28] that packs item in the lowest feasible position. The last one is unpacking heuristics. Since there are fewer heuristic algorithms for unpacking at present, we design two rules for unpacking. We define the positions of the items on the top layer of the bin (the first rule) and whose volume is less than a certain threshold (the second rule) as the valid positions.

To incorporate the heuristics into the DRL framework, we design two predictors to generate the binary feasibility mask $M$, indicating the valid and invalid actions. Only the actions that satisfy the heuristics are valid. Next, we use the feasibility masks to modulate the outputs of the packing and unpacking branches [29]. In our DRL framework, the actor network outputs the unnormalized scores $l$ (logits) which are then converted into an action probability distribution by a softmax operation. For the invalid actions, we replace their corresponding logits with a large negative value $-10^8$. Thus the probability of the invalid actions output by the softmax layer is virtually zero. We consider the action constraint as the renormalized probability distribution $\pi(a_t|s_t)$:

$$\pi(a_t|s_t) = softmax(f_{cons}(l_i)) = \begin{cases} \frac{\exp(l_i)}{\sum_j^N \exp(l_j)}, & M_i = True \\ 0, & M_i = False \end{cases} \tag{3}$$

where $N$ is the number of the valid actions, and

$$f_{cons}(l_i) = \begin{cases} l_i, & M_i = True \\ -10^8, & M_i = False \end{cases}.$$ (4)

**Reward function.** We design a step-wise reward for the bin packing task. It contains two parts: the volume utilization ratio $r_v$ and the wasted space ratio $r_w$. $r_v$ is defined as the ratio of the volume of the items contained in the bin to the total bin volume:

$$r_v = \frac{\sum_i^n l_i \times w_i \times h_i}{L \times W \times H}.$$ (5)

$r_w$ is defined as the ratio of the wasted space volume to the bin volume and denotes the free space where the items cannot be packed under the current bin state. Fig. 4 shows an example of the wasted spaces highlighted as the red dotted cuboids. We encourage the unpacking action to reduce the wasted spaces. The whole reward is defined as:

$$R_t(s_t, a, s_{t+1}) = r_v - \alpha r_w,$$ (6)

where $\alpha$ is a weighting parameter.

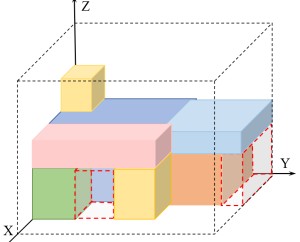

Figure 4: An example of the wasted spaces.

## 3.2 Implementation of Real-world Logistics Application

To validate the actual bin packing performance, we set up a real-world system to deploy the policy learned in the simulation.

**System configuration.** Fig. 5 shows the developed practical bin packing system. The UR5 robotic arm equipped with a suction gripper performs picking and placing actions. There are three workspaces: the conveyor belt for transporting items, the bin for holding the items and the buffer zone for temporarily storing items. RGB-D images of resolution $1280 \times 720$ are captured from two Intel RealSense D435i cameras mounted on two fixed tripods overlooking the conveyor belt and the bin.

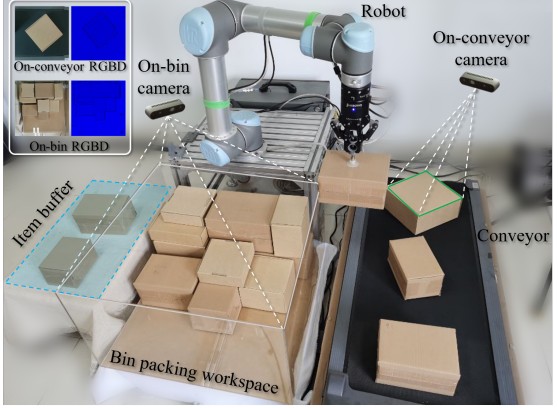

Figure 5: The developed bin packing system based on PUN.

**Sim-to-real transfer strategy.** Since the bin packing policy is learned in the simulated environment, we need a transfer strategy to apply it into the real world. The PUN takes as input two types of state representations, the bin heightmap and the item size map. To obtain the bin heightmap, we filter the observation captured by the camera and map it onto a discrete orthogonal grid using bilinear interpolation [30]. The mapped grid has the same resolution as the size of the bin (i.e. $L \times W \times H$). To obtain the item size map, we perform boundary detection on the image of the item on the conveyor belt and calculate its 3D sizes. To ensure the safety of the bin packing process, the item sizes are rounded up to discrete values as the final state representations.

# 4 Experimental Results

In this section, we conduct experiments and ablation studies to evaluate the proposed method. We also provide a supplementary video to demonstrate a real robotic system for the online 3D-BPP.

## 4.1 Data Generation and Implementation Details

To train and test the proposed network, we create three bin packing datasets based on previous works [13, 14], namely DATA-1, DATA-2 and DATA-3. To generate DATA-1, we cut a bin of size $S^d$ with

Table 1: Comparative results of different bin packing methods.

| Method | DATA-1 | | | DATA-2 | | | DATA-3 | | |
|---|---|---|---|---|---|---|---|---|---|
| | Uti. | Num. | Sta. | Uti. | Num. | Sta. | Uti. | Num. | Sta. |
| *Heuristic* | | | | | | | | | |
| Random | 0.363 | 15.66 | 0.131 | 0.348 | 9.61 | 0.134 | 0.366 | 10.16 | 0.127 |
| Column Building [31] | 0.629 | 25.75 | 0.125 | 0.566 | 14.76 | 0.124 | 0.571 | 15.26 | 0.127 |
| Floor Building [28] | 0.634 | 25.91 | 0.127 | 0.557 | 14.52 | 0.139 | 0.568 | 15.17 | 0.136 |
| First Fit [19] | 0.611 | 24.92 | 0.129 | 0.571 | 14.94 | 0.132 | 0.572 | 15.23 | 0.136 |
| Corner point [2] | 0.662 | 26.76 | 0.121 | 0.654 | 17.05 | 0.139 | 0.641 | 17.10 | 0.124 |
| Extreme Point [7] | 0.667 | 27.73 | 0.126 | 0.584 | 15.20 | 0.115 | 0.586 | 15.62 | 0.129 |
| Empty Maximal Space [8] | 0.669 | 27.79 | 0.114 | 0.652 | 16.99 | 0.118 | 0.649 | 17.31 | 0.124 |
| *Learning-based* | | | | | | | | | |
| Zhao et al. [13] | 0.687 | 28.61 | 0.103 | 0.632 | 16.39 | 0.114 | 0.642 | 16.97 | 0.111 |
| Yang et al. [14] | 0.704 | 29.13 | 0.097 | 0.667 | 17.31 | 0.104 | 0.675 | 18.05 | 0.105 |
| Zhao et al. [25] | 0.834 | 32.91 | 0.084 | 0.819 | 20.93 | 0.087 | 0.813 | 21.37 | 0.092 |
| Ours | **0.855** | **34.34** | **0.061** | **0.826** | **21.11** | **0.073** | **0.830** | **21.82** | **0.071** |

$d \in \{x, y, z\}$ into items of different sizes along the directions of length, width and height. The volume sum of all items is equal to the bin volume, and each item size $s^d \in \mathbb{Z}^+$ is not larger than $S^d/2$. The item sequence is randomly shuffled to increase data diversity. To generate DATA-2 and DATA-3, we first predefine 64 items of different sizes, and then cut a bin of size $S^d$ into different items with the predefined sizes. When cutting the bin into items, we record the size of each item and its position in the bin. Then the items are sorted into a new sequence according to different rules designed in [13]. Note that either DATA-2 or DATA-3 contains fewer items than DATA-1 while the average item volume is larger, which inevitably affects the performance.

In our implementation, the PUN is built based on actor-critic framework and we extend the framework to a two-branch architecture. We create 10 parallel virtual environments for collecting on-policy training samples. In each iteration, the packing and unpacking samples are used to update the packing and unpacking branches respectively. When one branch is active for gradient update, the other one is fixed and its gradient is set to 0. We train PUN for 10000 epochs and collect 5120 episodes in every epoch. Our models are developed using PyTorch and trained with an NVIDIA 2080Ti and an Intel i9-9900K CPU @ 3.60GHz. The hyperparameter settings that we used in the experiments are as follows: the discount factor $\gamma$ is 0.95, the learning rate $l_r$ is 0.003, the value function coefficient $\alpha$ is 0.5 and the entropy coefficient $\beta$ is 0.01.

## 4.2 Bin Packing Performance Evaluation

**Baseline Methods.** We compare the proposed method with 10 baseline methods for the online 3D-BPP, which can be classified into two groups. The first group contains a random packing policy and 6 heuristic methods. The second group contains 3 state-of-the-art DRL-based methods. For these DRL-based methods, the agent would terminate an episode when there is no available space for packing the current item in the bin. Note that the method proposed by Zhao et al. [25] has different variants and we select the one with the best performance as the baseline (i.e. PCT&EV in [25]).

**Results.** We use 3 metrics to evaluate the bin packing performance. The metric *Uti.* is the space utilization defined as the average volume ratio of all packed items to the bin. The metric *Num.* is the average number of all packed items. The metric *Sta.* is the standard deviation of *Uti.*

The comparative results reported in Table 1 show that our method outperforms all baseline methods in terms of all evaluation metrics. Such results indicate that PUN utilizes the bin space more sufficiently and performs more robustly than other competitors. Besides, due to the difference in the size and the number of the items in the 3 datasets as mentioned in Section 4.1, the bin packing methods performs generally better on DATA-1 than on DATA-2 and DATA-3. We also visualize the bin packing results for a qualitative comparison where we create 6 test sets by changing the item order of DATA-1. The visualization results in Fig. 6 show that our method outperforms the baselines.

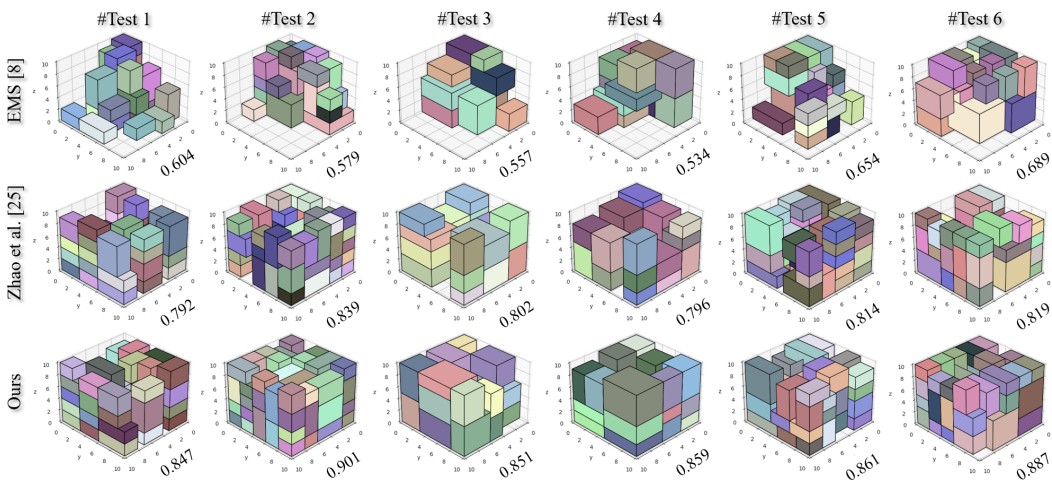

Figure 6: Visualization results of different methods. The values beside each sub-figure are the space utilization.

## 4.3 Ablation Studies

**Evaluation of PUN variants.** To demonstrate the designing choice of PUN, we create 3 variants for it and perform a series of ablation studies: **Packing-only** where only the packing branch of PUN is retained, **PUN-random-unpacking** where the unpacking branch of PUN randomly selects items to be unpacked, **PUN-no-constraint** where PUN is trained without heuristics-based constraints.

The comparative experiments are conducted on DATA-1, and the results are shown in Fig. 7, leading to several observations. First, PUN outperforms its variants in terms of both *Uti.* and *Num.*, indicating the effectiveness of the proposed network design. Second, compared to the variant without the unpacking action (Packing-only), the variants with the unpacking action (PUN, PUN-no-constraint, and PUN-random-

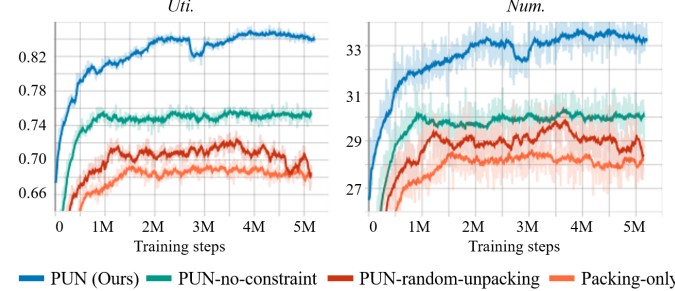

Figure 7: Learning curves of different PUN variants.

unpacking) achieve better performance in terms of both *Uti.* and *Num.*. Such results show that the introduction of the unpacking mechanism effectively improves the bin packing performance, which demonstrate the proposed idea of introducing the unpacking action for online 3D-BPP. Third, for the 3 variants with the unpacking action, the learning-based ones (PUN and PUN-no-constraint) perform better than the one based on the random policy (PUN-random-unpacking). It demonstrates the significance of the proposed DRL framework for learning the synergies between packing and unpacking. Finally, PUN outperforms the variant without any heuristic constraint (PUN-no-constraint), which indicates that the heuristics-based action constraints provides useful guidance for the training of PUN and improves the bin packing performance.

**Exploration of different reward functions.** To investigate the effect of different settings of the reward function, we design 3 reward functions and compare them with the proposed one on DATA-1. The first one is a constant value reward, defined as 1 if the packing action is successfully performed, 0.5 if the unpacking action is successfully performed, and 0 otherwise. The second reward is the final volume ratio of

Table 2: Comparison of different reward functions

| Reward | Uti. | Num. | Sta. |
|---|---|---|---|
| Constant reward | 0.742 | 29.96 | 0.085 |
| Final-vol reward | 0.745 | 30.07 | 0.091 |
| Item-num reward | 0.771 | 31.84 | 0.087 |
| Our reward | 0.855 | 34.34 | 0.061 |

all packed items to the bin, which we record after all steps and assign to each step of bin packing.

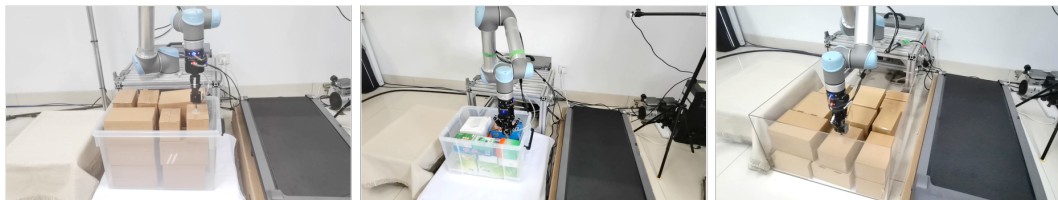

Figure 8: Results of the real-world bin packing tests produced by our method.

Table 3: Effect of different buffer capacities

| Buffer capacity | 2 | 4 | 6 | 8 | 10 | 12 | 14 | No limit |
|---|---|---|---|---|---|---|---|---|
| Uti. | 0.739 | 0.791 | 0.815 | 0.834 | 0.852 | 0.855 | 0.854 | 0.855 |
| Num. | 29.21 | 30.74 | 32.57 | 33.19 | 34.21 | 34.32 | 34.31 | 34.34 |

The last one is the item number reward, defined as the number of the packed items in the bin. According to the results listed in Table 2, our method outperforms the competing methods in terms of the space utilization and the number of packed items, which demonstrates the effectiveness of the designed reward.

**Exploration of buffer capacity.** We conduct the ablation study on DATA-1 to investigate the effect of different capacity $B$ of the item buffer. For this experiment, the terminal condition of an episode is that the number of items in the buffer zone exceeds the buffer capacity or there is no position for packing and unpacking in the bin. The results reported in Table 3 show that a larger $B$ results in better bin packing performance. However, when the buffer capacity exceeds 12, there is no significant performance gain.

### 4.4  Real-World Bin Packing Demonstration

We also demonstrate the proposed method in a real logistics scenario where the implementation details can be found in Section 3.2. In this task, the robotic arm packs items from the conveyor belt into the bin one by one through the learned synergies between packing and unpacking. Fig. 8 shows some testing results produced by our method. It can be seen that various items are packed into different bins with high space utilization. The full bin packing process can be found in the supplementary video, which demonstrates that our method reliably completes the real-world online 3D bin packing task in different logistics scenarios.

### 4.5  Limitation

Due to the measurement error of the vision sensors, the estimated item size and bin heightmap may have some deviations. The policy takes as input inaccurate state observations of the scene and consequently predicts inappropriate packing or unpacking actions. A possible solution is to introduce a "re-packing" strategy to reposition the inappropriately packed/unpacked items. Besides, the proposed method focuses mainly on packing cuboid items while it cannot handle the irregularly shaped items. This is because the state representations of PUN are unable to precisely represent the 3D geometry of irregularly shaped items. Consequently, an irregularly shaped item will be regarded as the cuboid one which tightly circumscribes it by PUN. Thus, future work is to extend PUN to the items with diverse shapes by designing more powerful state representations of the item and the bin.

## 5  Conclusion

We introduce the idea of unpacking for the online 3D-BPP. Based on it, we propose the packing-and-unpacking network (PUN) which learns the synergies between packing and unpacking actions via DRL. We also incorporate the heuristics which formulate the packing and unpacking experiences of human packers into PUN to further improve the bin packing performance. The results of both simulated and real-world experiments demonstrate the effectiveness of the proposed method.

**Acknowledgments**

This work was supported in part by the National Key Research and Development Plan of China under Grant 2021ZD0112002 and Grant 2018AAA0102504, in part by the National Natural Science Foundation of China under Grant 61991411 and Grant U1913204, in part by the Natural Science Foundation of Shandong Province for Distinguished Young Scholars under Grant ZR2020JQ29, and in part by Project for Self-Developed Innovation Team of Jinan City under Grant 2021GXRC038.

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
