# OpenReview forum: "Towards Online 3D Bin Packing: Learning Synergies between Packing and Unpacking via DRL"
_robot-learning.org/CoRL/2022/Conference — CoRL 2022 Poster_

### Official Review · Reviewer_Wn5v · 2022-07-29

**Originality:** Very Good
**Technical Quality:** Good
**Clarity Of Presentation:** Good
**Impact:** 3

**Recommendation:**

Weak Accept: I recommend accepting the paper, but will not argue for my recommendation if the majority of other reviewers have a different opinion.

**Summary:**

The author presented a 3D bin packing algorithm, where unpacking action can be used to correct wrong decisions. They propose a packing-and-unpacking network (PUN) based on Deep Reinforcement Learning (DRL). The architecture is composed of two networks, a first one for the packing action and the second one for the unpacking one. At execution, both network output the value function evaluated at the current state and the action with the max reward is chosen. The networks output also the packing and unpacking position, respectively. The proposed algorithm is tested against 10 benchmarks both in simulation and experiments.

**Issues:**

I think that the clarity of the paper can be improved. In the Method section, I would add a training subsection where the authors could explain better the training process. It seems that the value function associated with the packing policy is a function of how good the unpacking policy is, i.e., the problem cannot be decoupled. Are the authors considering some coupling effect in the training process? It seems that it is a good idea to decompose the problem into subtasks, but could the authors provide a discussion on the pro and cons?

**Quality Of The Limitations Section:**

Limitations are addressed clearly

**Reviewer Expertise:**

4: The reviewer is confident but not absolutely certain that the evaluation is correct

**Robotics Focus:**

Sufficient demonstration on hardware

**Strengths And Weaknesses:**

-Strengths
1) The experiments clearly show the effectiveness of the proposed strategy both in simulation and experiments.
2) The idea of having an unpacking option is useful in practice.

-Weaknesses
1) It is a bit unclear how the two networks are trained.
2) The algorithm is not theoretically sound as it is based on several heuristics.

**Summary Of Recommendation:**

I think that overall the Strengths are more than the Weaknesses. In particular, I think that the result section clearly shows the benefit of the method. My only recommendation is to better explain the training process for clarity, and I think that such change is in the reach of the authors.

---

> ### Author Response · Authors · 2022-08-22
> **Response to Reviewer Wn5v**
>
> Thanks for the reviewer’s constructive and positive feedback. We are encouraged that the reviewer acknowledged the effectiveness and practicability of our approach. Our responses to the reviewer’s concerns are given below:
>
> ##### **Q4.1: Training process details**
>
> A4.1: We shall add a subsection in the final version of the paper to describe the training process. Specifically, PUN is built based on PPO and we extend it to a two-branch architecture. We create 10 parallel virtual environments for collecting on-policy training samples. In each iteration, the sampled packing and unpacking data are used to update the packing and unpacking branches respectively. When one branch is active for gradient update, the other one is fixed and its gradient is set to 0. We train PUN for 10000 epochs and collect 5120 episodes in every epoch. Our models are developed using PyTorch and trained with an NVIDIA 2080Ti and an Intel i9-9900K CPU at 3.60GHz. It takes about 6 hours to train our model for 10000 epochs. The hyperparameter settings that we used in the experiments are shown in the table below:
>
> | Parameters  | Value |
> | ------------ | ------------ |
> | Learning rate  |  0.003 |
> | Discounted factor   | 0.95  |
> | Weight for entropy loss  | 0.003 |
> |  Weight for value loss | 0.5  |
> |  Clip epsilon | 0.2  |
> |  Max grad norm |  0.5 |
> |  GAE lambda | 0.95  |
>
> ##### **Q4.2: Joint training vs. separate training**
>
> A4.2: Our bin packing task is a multi-step decision problem as the early packing or unpacking actions will affect the decision and execution of subsequent actions. Thus it is reasonable to integrate packing and unpacking into a whole framework for an uniform learning objective (e.g. higher space utilization). If we decompose the learning task into two subtasks (packing and unpacking) and perform the training separately, the relevance between packing and unpacking would be ignored. In other words, a separately trained unpacking model cannot determine whether the executed unpacking actions are useful for packing more items into the bin. As a result, when we train the unpacking model separately, it is difficult to provide effective signals to guide the learning. Moreover, in the inference stage, we must combine the two separately trained parts (packing and unpacking) as whole, which is a very challenging task as we need to additionally design a reasonable “selection module” to select the action type (packing or unpacking) for execution.

---

### Official Review · Reviewer_WbWp · 2022-07-30

**Originality:** Good
**Technical Quality:** Very Good
**Clarity Of Presentation:** Very Good
**Impact:** 4

**Recommendation:**

Weak Accept: I recommend accepting the paper, but will not argue for my recommendation if the majority of other reviewers have a different opinion.

**Summary:**

The authors propose a deep RL based architecture for the 3D bin packing problem (3D BPP). The policy is learnt in simulation and is able to capture the synergy between picking and packing to achieve a target 3D packing configuration that minimizes the wasted space in the box. The policy takes in as input a heightmap and state representation specifying the item size and outputs pixel-locations for the pack/unpacking actions. The policy learns the synergy between pick and unpacking actions. Apart from outperforming prior learning and heuristic methods in simulation, the policy transfers to the real robot and even everyday objects (like cereal boxes).


**Issues:**

 See above.

**Quality Of The Limitations Section:**

Limitations are addressed clearly

**Reviewer Expertise:**

5: The reviewer is absolutely certain that the evaluation is correct and very familiar with the relevant literature

**Robotics Focus:**

Sufficient demonstration on hardware

**Strengths And Weaknesses:**

Strengths:
The paper is well written. It also has good ablations and experimental comparisons with prior work. I’m not familiar with the SOTA of the field, but the task seems unique enough.
The approach is also demonstrated on the real robot with strong sim2real transfer results.

Weaknesses:
The data generation process could be explained better.
The approach only seems to work on limited object geometries and fixed-size box volumes. It would be great to clarify if the method can be expanded or if it's beyond the scope.


**Summary Of Recommendation:**

 See above.

---

> ### Author Response · Authors · 2022-08-22
> **Response to Reviewer WbWp**
>
> Thanks for the reviewer's thorough review and helpful suggestions. We are encouraged that the reviewer recognized our paper's good writing, uniqueness, adequate experiments and strong sim2real transfer results. Our detailed responses to the reviewer's concerns are given below:
>
> ##### **Q3.1: The data generation process**
>
> A3.1: We shall adopt the reviewer's suggestion and add more details in Section 4.1 to clearly describe the data generation process. Specifically, we create three different bin packing datasets based on previous works [13, 14], namely DATA-1, DATA-2 and DATA-3. The data generation process for the three datasets are described as follows:
>
> **DATA-1:** For a bin of size $S^d$ with $d \in (x,y,z)$ , we cut it into items of different sizes along the directions of length, width, and height. The volume sum of all items is equal to the bin volume, and each item size $s^d \in Z^+$ is not larger than $S^{d}/2$. In total, DATA-1 contains 20000 item sequences and we randomly shuffle the item sequence in every training episode to increase data diversity.
>
> **DATA-2 and DATA-3:** Following Zhao et al. [13], we first predefine 64 items of different sizes, and then cut a bin of size $S^d$ into different items with the predefined sizes. When cutting the bin into items, we record the size of each item and its position in the bin. Then, the items are sorted into a new sequence according to different rules to generate DATA-2 and DATA-3. For DATA-2, we sort the items in the sequence based on *Z* coordinates of their left-front-bottom corner position in a bottom-to-top manner. For DATA-3, the cut items are sorted based on their stacking dependency: an item can be added to the sequence only after all of its supporting items are there. In total, DATA-2 and DATA-3 both contain 10000 item sequences.
>
> ##### **Q3.2: The scalability over object geometry and size**
> A3.2: As described in the limitation section (Section 4.5), our method currently focuses mainly on packing cuboid items while it cannot handle the irregularly shaped items. In the future, we aim to extend the proposed method to the items with diverse shapes by designing more powerful state representations of the item and the bin. Besides, to explore the scalability over item size, we create a new dataset containing 343 different sizes of items and run tests on it. Note that the model is trained on DATA-1 that contains 64 different sizes of items while tested on the newly created dataset. The results shown below indicate that our method has a good generalization ability on unseen items and works well on a wide range of item sizes. Even so, the bin packing performance would drop slightly when the model is tested using the data with unseen items.
>
> |  Method | Uti.  |  Num. |  Sta. |
> | :------------: | :------------: | :------------: | :------------: |
> | PUN  | 0.803  | 23.82  |  0.081 |

---

### Official Review · Reviewer_axHb · 2022-07-30

**Originality:** Good
**Technical Quality:** Good
**Clarity Of Presentation:** Excellent
**Impact:** 3

**Recommendation:**

Weak Reject: I recommend rejecting the paper, but will not argue for my recommendation if the majority of other reviewers have a different opinion.

**Summary:**

The paper presents a solution to the 3D Bin Packing problem via the use of Deep Reinforcement Learning. Specifically, a 2 branch architecture is proposed to introduce the unpacking action and to learn how to apply optimally both packing and unpacking actions on boxes off of a conveyor belt in order to fill a bin while leaving as little space empty as possible.

**Issues:**

Ln. 88: discretize? Or define?

Ln. 91-92: "The state of the current item n with the size of 91 ln × wn × hn is represented as..." That means that the state of each object never changes then since no object will ever change in width, height and length? If I'm misinterpreting the sentence, please rephrase to clarify.

Ln. 98: nit: tiny typo "defined".

Ln. 110: "a shared feature extraction module": how does this look like?

Ln. 112: "Either the packing or the unpacking branch consists of...": Did authors mean to say "The packing and unpacking branches each consist of..."?

Ln. 118. "based on the heuristic rules": what are the heuristic rules? Assuming they are explained later in the paper, however please add such note here to point the reader to that fact, e.g. "based on heuristic rules that will be explained in section XXX".

Ln 132-133: "The agent will continue to predict the packing or unpacking action for the item n at the next time step until packing it into the bin" --> how does this avoid possible infinite loops?

General comment: I'm assuming that based on the items and the capacity of the temporary buffer, there can be no solutions found. Does the method simply stop? Or does a best effort as far it's possible? Or?

Ln. 147: "that on the top". Typo.

Ln. 147-148: a) Is this the first rule for unpacking? If yes, please explain what it's aiming to do. b) Was the second rule omitted? Or mentioned later?

Equations 3 & 4: what is M_i?

Ln. 251-254: Not sure what's the rationale for reward function 1. Success in packing or unpacking actions alone does not really give any indication about whether that was a good move. Please explain.

Ln. 64-65: I assume that the no gains observed above 12 was due to the number of items not changing. It's not obvious why this would still hold if number of items on the conveyor belt was very large. Have you increased the number of items and still observed this? If yes, do you hypothesize on why?

General comment: In general, the idea of unpacking and "retrying" in assembly tasks is not new. There is a whole line of work by Zeng et. al on Assembly & Disassembly (Transporter Nets, Form2Fit etc.) that was not cited where authors using Imitation Learning techniques to solve assembly tasks.

Also, have the authors attempted to solve this problem with Imitation Learning techniques like Behavioral Cloning? Not convinced that simple BC wouldn't have comparable results. Would like to have seen this at least as a baseline.


**Quality Of The Limitations Section:**

Limitations are addressed clearly

**Reviewer Expertise:**

5: The reviewer is absolutely certain that the evaluation is correct and very familiar with the relevant literature

**Robotics Focus:**

Sufficient demonstration on hardware

**Strengths And Weaknesses:**

Strengths: Cool method, very well presented, enjoyable to read. Evaluation was quite thorough.

Weaknesses: While evaluation was thorough, would have liked to have seen simple Imitation Learning approaches discussed, cited and compared against (see below for more details).

**Summary Of Recommendation:**

Recommendation is based on not having cited, discussed and compared against at least one simple Imitation Learning approach. Imitation Learning approaches where expert demonstrations have both packing and unpacking actions, should be able to do fairly well on the task which would at least make a strong argument in favor of the efficacy of the approach if it outperformed them. I am willing to change my recommendation if authors address this adequately during rebuttal period.

---

> ### Author Response · Authors · 2022-08-22
> **Response to Reviewer axHb (1/3)**
>
> Thanks for the reviewer's thoughtful review and constructive feedback. We appreciate the reviewer's acknowledgment of our paper in terms of both the idea and the structure and are glad that the reviewer enjoys reading our paper. Detailed responses to the concerns are as follows:
>
> ##### **Q2.1: Imitation learning approaches for 3D bin packing**
> A2.1: As suggested by the reviewer, we have made a comparison with a popular imitation learning approach, namely Behavioral Cloning (BC) [A, B]. First, we collect the expert demonstrations of online 3D bin packing including the state observations and corresponding packing/unpacking actions. Then, we train a network to minimize the differences between the learned bin packing policy and the expert bin packing policy. To collect the expert demonstrations, we develop an interactive interface, by which the human can complete the online 3D bin packing task through packing or unpacking operations. Then we ask 10 human users (all are master and PhD students with science or engineering backgrounds) to implement the bin packing tasks and record the samples (state and action). In total, the collected data contain 1000 expert demonstrations. We train the network with such expert demonstrations and show the experimental results in the table below. It can be seen that the BC baseline achieves moderate bin packing performance and more expert demonstrations may bring more performance gains while large-scale data collection is difficult. We shall add the BC baseline in Table 1 in the final version of the paper.
>
> |  Method |  Uti. | Num.  | Sta.  |
> | ------------ | ------------ | ------------ | ------------ |
> |  Behavioral Cloning method | 0.697  | 28.57  |  0.114 |
>
> **Reference:**
>
> [A] Bain M, Sammut C. "A Framework for Behavioural Cloning". Machine Intelligence 15. 1995: 103-129.
>
> [B] Ross S, Bagnell D. "Efficient reductions for imitation learning". Proceedings of the thirteenth international conference on artificial intelligence and statistics, 2010: 661-668.
>
> ##### **Q2.2: Difference from the work of Zeng et al. on Assembly&Disassembly**
> A2.2: Thanks for pointing out the missing references. In general, the work of Zeng et al. on Assembly&Disassembly (Transporter Nets [C], Form2Fit [D] etc.) focus on robotic assembly and formulate the assembly task as a shape matching problem. Essentially, the core of assembly task is to accurately place objects in target locations. While in real-world 3D bin packing task, accurately placing items into bin is a relatively easier part and performs robustly in most cases. Instead, learning an optimal bin packing policy that maximizes the space utilization is the crucial part of online 3D bin packing and has not been well addressed. In this light, we present an effective approach for online 3D bin packing and demonstrate the state-of-the-art bin packing performance.
>
> What is more important, the “unpacking” or “retrying” strategies employed in the work of Zeng et al. are quite different from our unpacking mechanism. They just used the “unpacking” actions to disassemble the kits to acquire training data while do not learn how to unpack and neglect the mutual influence between “packing” and “unpacking”. In contrast, we proposed to improve the bin packing performance with the synergies between packing and unpacking and presented a novel framework to learn the optimal synergistic policy. Even so, the work of Zeng et. al demonstrated the applicability and effectiveness of solving the assembly and disassembly tasks using imitation learning techniques, offering a new perspective to solve the online 3D-BPP. We shall cite the related work (Transporter Nets, Form2Fit etc.) and add corresponding discussions in the final version of the paper.
>
> **Reference:**
>
> [C] Zeng, A, Florence P, et al. "Transporter Networks: Rearranging the Visual World for Robotic Manipulation". Proceedings of the 2020 Conference on Robot Learning.
>
> [D] Zakka K, Zeng A, et al. "Form2fit: Learning shape priors for generalizable assembly from disassembly". 2020 IEEE International Conference on Robotics and Automation (ICRA). IEEE, 2020: 9404-9410.
>
> ##### **Q1.3: Other issues**
> > Ln. 88: discretize? Or define?
>
> A: Yes, it should be “discretize”.
>
> > Ln. 91-92: “The state of the current item n with the size of ln * wn * hn is represented as” That means that the state of each object never changes then since no object will ever change in width, height and length? If I'm misinterpreting the sentence, please rephrase to clarify.
>
> A: Yes, the state representation of each object to be packed is fixed and the objects with the same size have the same state representations. To determine where to place a coming item, the agent must know the geometric information of the arrived item and the stacked items in the bin. Hence we use the heightmap to represent the state of the bin and use the size map to represent the state of the current item.

---

> ### Author Response · Authors · 2022-08-22
> **Response to Reviewer axHb (2/3)**
>
> > Ln. 98: nit: tiny typo "defined".
>
> A: Sorry for the typo. We shall correct it in the final version of the paper.
>
> > Ln. 110: "a shared feature extraction module": how does this look like?
>
> A: The feature extractor is implemented with 5 fully-connected layers. The heightmap and size map are flattened into one-dimension vectors which are then concatenated together and forwarded into the feature extraction module. We shall add such details in Figure 2 in the final version of the paper.
>
> > Ln. 112: "Either the packing or the unpacking branch consists of...": Did authors mean to say "The packing and unpacking branches each consist of..."?
>
> A: Exactly! Sorry for the unclear description and we shall correct it in the final version of the paper.
>
> > Ln. 118. "based on the heuristic rules": what are the heuristic rules? Assuming they are explained later in the paper, however please add such note here to point the reader to that fact, e.g. "based on heuristic rules that will be explained in section XXX".
>
> A: The heuristic rules mentioned here are physics heuristics, packing heuristics and unpacking heuristics. We provide the detailed descriptions of these heuristics in the “Heuristics-based action constraint” part of Section 3.1. And as suggested, we add such note here to guide the reader to catch the details easily.
>
> > Ln 132-133: "The agent will continue to predict the packing or unpacking action for the item n at the next time step until packing it into the bin" --> how does this avoid possible infinite loops?
>
> A: In our early studies, we found that the training process often traps into infinite loops between packing and unpacking. To avoid such cases, we set definite termination conditions for a training episode: the agent will terminate the current training episode if there is no available space for packing the current item into the bin and it cannot find an unpackable item from the bin. In other words, the current training episode must stop if the packing and unpacking actions are both unavailable. Note that in our setting, not all items packed in the bin are unpackable. In the “Heuristics-based action constraint” part of Section 3.1, we design an unpacking heuristic module to define which items are allowed for unpacking. Briefly, an unpackable item is on the top layer of the stacked items and its volume must be smaller than the current item to be packed. In such a manner, the problem of infinite loops is totally avoided.
>
> > General comment: I'm assuming that based on the items and the capacity of the temporary buffer, there can be no solutions found. Does the method simply stop? Or does a best effort as far it's possible? Or?
>
> A: Yes. Currently, our method will stop the bin packing process if the packing and unpacking actions are both unavailable. For the packing action, it cannot be executed anymore if there is no available space for packing the current item. For the unpacking action, it cannot be executed if there is no unpackable position in the bin or the temporary buffer is full.
>
> > Ln. 147: "that on the top". Typo.
>
> A: Sorry for the typo. We shall correct it in the final version.
>
> > Ln. 147-148: a) Is this the first rule for unpacking? If yes, please explain what it's aiming to do. b) Was the second rule omitted? Or mentioned later?
>
> A: Sorry for our unclear description which might cause a misunderstanding. The sentence stated here actually describes two different unpacking rules. The first one is that an unpackable item should be on the top layer of the stacked items. This rule imposes realistic physical constraint to the unpacking actions (i.e., unpacking an item packed inside the stacked item pile obviously defies the physics logic). The second one is that the volume of an unpackable item must be smaller than the current item to be packed. We design this rule to guarantee a positive volume exchange (i.e. unpacking a small item to save spaces for packing a big one). Besides, the possible infinite loops during training can be avoided with such a rule. We shall rewrite the descriptions of the two unpacking heuristic rules in the final version of the paper.

---

> ### Author Response · Authors · 2022-08-22
> **Response to Reviewer axHb (3/3)**
>
> > Equations 3 & 4: what is M_i?
>
> A: $M_i$ denotes the binary feasibility mask, indicating the valid and invalid actions. The details about the feasibility mask are provided in the “Heuristics-based action constraint” part of Section 3.1.
>
> > Ln. 251-254: Not sure what's the rationale for reward function 1. Success in packing or unpacking actions alone does not really give any indication about whether that was a good move. Please explain.
>
> A: Yes, this reward function does not give any indication about whether that was a good move. In essence, the constant reward is a survival encouragement strategy. For each step, no matter good or bad (i.e. beneficial for overall utilization improvement or not), we both give a constant positive reward to encourage the agent to perform longer in an episode. Intuitively, longer action steps allow the agent to pack more items and thus result in higher space utilization. However, the comparative results shown in Table 2 in the paper demonstrate that such a reward function does not work well.
>
> > Ln. 64-65: I assume that the no gains observed above 12 was due to the number of items not changing. It's not obvious why this would still hold if number of items on the conveyor belt was very large. Have you increased the number of items and still observed this? If yes, do you hypothesize on why?
>
> A: As the reviewer suggested, we conduct additional experiments to explore the buffer capacity *C* for the best performance. Briefly, we experimentally found that the item number increase does not change *C* while the larger differences of item sizes lead to a larger *C*. Specifically, we create two datasets and perform comparative experiments. The first one is named Data-num. We use the data generation method of DATA-1 to create training item sets and then merge two separate sets as a new one. As such, the item number of a new item set is obviously larger. The second one is named Data-size. We predefine 216 items with larger size difference than that of Data-num and use them to create the item sets. For fair comparison, we keep the item number of the two datasets as the same. The experimental results shown in the table below demonstrate the above findings. In the cases with larger size difference, it often happens that the current item is too large to be packed into the bin. Then the policy needs to unpack more items out of the bin before the current item is packed into the bin, which naturally requires a larger buffer capacity.
>
> | Dataset  | Data-num  | Data-size  |
> | :------------: | :------------: | :------------: |
> | Buffer capacity *C* |  12 | 16  |

---

### Official Review · Reviewer_G14Q · 2022-08-02

**Originality:** Excellent
**Technical Quality:** Fair
**Clarity Of Presentation:** Excellent
**Impact:** 4

**Recommendation:**

Weak Accept: I recommend accepting the paper, but will not argue for my recommendation if the majority of other reviewers have a different opinion.

**Summary:**

The paper presents a novel deep-RL based method for the online 3D bin packing problem. Here, cuboidal items arrive in random order and need to be packed as tightly as possible. The key idea of this paper is to enable to robot to "backtrack" suboptimal packing actions by temporarily unpacking part of the stack into a buffer region, and thus build more optimal packing configurations. The method is evaluated on extensive simulation experiments and a real world demonstrator.


**Issues:**

- Metrics/evaluation: Measure efficiency in terms of number of (additional unpacking) actions
- Fair baseline comparison: Is PUN-packing-only significantly worse than Zhao et al? If yes, how much better is Zhao et al with some simple (e.g. random) or clever (?) unpacking strategy?
- Minor issues

(see Strengths and Weaknesses section for details)

**Quality Of The Limitations Section:**

Additional details required

**Reviewer Expertise:**

5: The reviewer is absolutely certain that the evaluation is correct and very familiar with the relevant literature

**Robotics Focus:**

Sufficient demonstration on hardware

**Strengths And Weaknesses:**

## Strengths

The paper is extremely well-written and easy to understand, despite the technical intricacies of the work. The method is technically sound, offering a great mix of explicit modelling and learning. The experiments are thoughtfully conceived, carried out and analyzed. The method demonstrates performance improvements over the state of the art and purely heuristic methods, mainly owed to the ability to perform backtracking through unpacking. Overall the paper was a very enjoyable and insightful read.

## Weaknesses

I have two concerns regarding the evaluation of the work.

### Efficiency in terms of number of actions

The paper fully focuses on the cube utilization, i.e. reducing wasted space. While this is obviously a key metric, in industrial applications efficiency and units per hour (UPH) are equally important factors -- and obviously there is a trade-off between cube utilization and efficiency. I'm missing an analysis on how many unpacking actions are required to reach the improved cube utilization. This should be addressed by introducing an additional metric such as the average number of packing+unpacking actions.

### Unfair baseline comparison?

Directly related to the previous question is whether the baselines are evaluated on a fair basis. If I understand correctly, no baseline (except for PUN-random-unpacking) actually does unpacking. If I interpret Figure 7 correctly, PUN-packing-only (Uti. = ~0.67) significantly underperforms the best performing baseline Zhao et al. (Uti. = 0.834). This immediately raises the question how much better Zhao et al. would perform if it was extended to use unpacking. For example, how well would Zhao et al. perform with random unpacking?

### Minor points

- Heuristics (140-148). I would encourage the authors to make the paper self-sufficient and explain the heuristics, at least 1/2 sentence per heuristic, rather than just naming them.
- Figure 2. I was initially confused what "packing mask" and "unpacking mask" meant - I would suggest to explicitly mention "heuristics" in the figure to ensure to make clear that the Heuristics section 140-148 will explain it.
- L154: Typo, -e-8 is not a mathematical expression, better use $10^{-8}$
- L161: What does the t in the sum refer to? Is this a typo?
- Sim-to-real strategy. I don't fully understand how individual items are identified in the real world demo, isn't some sort of instance segmentation both on the converyor as well as on the stack needed to identify individual boxes?
- Limitations. I would suggest to discuss how perception and action errors might affect


**Summary Of Recommendation:**

The paper makes an important contribution to a highly relevant problem. The idea to introduce unpacking and learn an unpacking policy is compelling. However, the impact of unpacking needs to be analyzed more carefully, in particular on the efficiency of the approach (number of actions), and also on whether the baselines comparisons are fair.

---

> ### Author Response · Authors · 2022-08-22
> **Response to Reviewer G14Q (1/2)**
>
> Thanks for the reviewer’s thorough review and valuable feedback. We are encouraged that the reviewer acknowledged our idea and found our paper a very enjoyable and insightful read. Below are our responses to the concerns:
>
> ##### **Q1.1: Efficiency in terms of number of actions**
>
> A1.1: We agree with the reviewer that in addition to space utilization, efficiency is also important in bin packing applications. Thus we have conducted additional experiments using efficiency metrics. Specifically, we run 10,000 tests on DATA-1 using randomly generated samples and recorded in each test the final space utilization, the packed item number, the unpacking action number, and the sum of packing and unpacking actions. The average performance is reported in the table below. It can be seen that our approach achieves a high space utilization of 85.5% at the cost of about 20 unpacking actions and 74 total actions (packing+unpacking). In practice, we can set a top limit *N* of unpacking actions (the maximum allowed unpacking actions) to make a balance between utilization and efficiency. For example, we can set a small *N* in efficiency-sensitive applications while a large *N* in performance-sensitive applications.
>
> | Method  | &nbsp;&nbsp;&nbsp;Uti.&nbsp;&nbsp;&nbsp;  |  &nbsp;&nbsp;&nbsp;Num.&nbsp;&nbsp;&nbsp; | Unpacking action number  | Total action number  |
> | :------------: | :------------: | :------------: | :------------: | :------------: |
> |  PUN | 0.855  | 34.34 |  20.03 |  74.40 |
>
> ##### **Q1.2: Unfair baseline comparison**
> A1.2: In our paper, we implement PUN by incorporating the unpacking mechanism into a simple DRL baseline to clearly demonstrate the effectiveness of the unpacking mechanism. Actually, our unpacking mechanism has good generality and can be integrated with different packing-only baselines. To demonstrate this, we follow the reviewer’s suggestion and extend Zhao et al. [25] with random unpacking. Besides, we also attempt to design a simple rule to extend Zhao et al. with unpacking strategy. At the initial stage, the network performs the packing actions as usual. When an arrived item cannot be packed in the bin, the unpacking process is activated. For the unpacking process, we define the items on the top layer of the bin with volume less than a certain threshold as candidate items to be packed, and select the item with smallest volume to unpack. Then we conducted comparative experiments on DATA-1 and the results are given in the table below. It can be seen that compared to Zhao et al., the random unpacking leads to a slightly better bin packing performance and the rule-based unpacking brings a significant performance gain.
>
> | Method  | Uti.  | Num.  | Sta. |
> | ------------ | ------------ | ------------ | ------------ |
> | Zhao et al.  | 0.834  |  32.91 | 0.084  |
> | Zhao et al. + Random unpacking  | 0.840  | 33.14  | 0.080  |
> |  Zhao et al. + Rule-based unpacking | 0.852  |  34.03 |  0.068 |
>
> ##### **Q1.3: Minor points**
> A1.3: Thanks for pointing out these minor issues to improve the readability and enhance the rigor of our paper. We shall sort them out in the final version of the paper by adding the additional explanations, listed below:
>
> > Heuristics (140-148). I would encourage the authors to make the paper self-sufficient and explain the heuristics, at least 1/2 sentence per heuristic, rather than just naming them.
>
> A: The used heuristics are described as follows. The **Extreme Point** algorithm calculates all extreme points in the bin as candidate packing positions and randomly chooses a position to pack item. The **Empty Maximal Space** algorithm packs item in the position that can produce the largest empty orthogonal spaces in the bin. The **First Fit** algorithm packs item in the first position among all feasible positions. The **Floor Building** algorithm packs item in the lowest position among all feasible positions.
>
> > Figure 2. I was initially confused what "packing mask" and "unpacking mask" meant - I would suggest to explicitly mention "heuristics" in the figure to ensure to make clear that the Heuristics section 140-148 will explain it.
>
> A: We shall modify the “packing-mask” and “unpacking-mask” into “packing-heuristics” and “unpacking-heuristics” in Figure 2 and update the related descriptions accordingly in the Heuristics section.
>
> > L154: Typo, -e-8 is not a mathematical expression, better use $ 10^{-8}$
>
> A: We shall change -e-8 to a mathematical expression $ 10^{-8}$ .
>
> > L161: What does the t in the sum refer to? Is this a typo?
>
> A: '*t*' is indeed a typo and we shall update Eq. (5) by replacing '*t*' with '*n*', where *n* denotes the number of items contained in the bin.

---

> ### Author Response · Authors · 2022-08-22
> **Response to Reviewer G14Q (2/2)**
>
> > Sim-to-real strategy. I don't fully understand how individual items are identified in the real world demo, isn't some sort of instance segmentation both on the converyor as well as on the stack needed to identify individual boxes?
>
> A: In the real-world demo, we obtain the size of the upcoming item on the conveyor and the heightmap of the stacked items in the bin. Given an arrived item on the conveyor, we capture its RGBD images and calculate its 3D size via segmentation. For the bin heightmap, we do not need to segment and identify each item in the bin. Instead, we map the depth image of the bin to a discrete orthogonal grid according to the depth value to obtain the bin heightmap.
>
> > Limitations. I would suggest to discuss how perception and action errors might affect
>
> A: We shall also discuss the perception and action errors as the limitation of our work. Due to the measurement error of the vision sensors, the estimated item size and bin heightmap may have some deviations. As a result, the policy takes as input inaccurate state observations of the scene and consequently predicts inappropriate packing/unpacking positions, which may decrease the overall space utilization. In practice, we cannot guarantee that all predicted actions can be executed accurately as expected. For example, the placed item may collide with other items of the bin, resulting in a failed placing manipulation. Such action error would interrupt the bin packing process.

---

### Meta-Review · Area_Chair_PEWj · 2022-08-14

**Recommendation:** Accept (Poster)
**Confidence:** 4

**Metareview:**

This paper proposes a novel DRL based approach for the online 3D bin-packing problem. The main idea is inspired by how a human operator would have to use unpacking as an support action to backtrack sub-optimal packing actions. The synergies between packing and unpacking actions are optimized via DRL. The proposed method has several strengths: technical sound, a great mix of explicit modelling and learning, thoughtfully conceived, carried out and analyzed experiments, and demonstration on the real robot with strong sim2real. On the other hand, there are concerns about the used metrics/evaluation, i.e. can there be a trade-off between cube utilization and efficiency? In addition, there is a question on the choice of the baseline for comparisons where the PUN-packing-only baseline significantly underperforms the best performing baseline Zhao et al. (Uti. = 0.834). As another baseline, imitation Learning approaches can also be alternative as the authors have also discussed about the human performance baseline. For clarify, it's required to further explain about the data generation process and the training of the two proposed networks.

---------------

The authors have made a great rebuttal response to address the reviewers' concerns. Additional results and experiments are convincing. The authors should add them to the final version.


**Best Paper Nomination:**

No

---

> ### Author Response · Authors · 2022-08-22
> **Response to Area Chair PEWj**
>
> We thank the area chair and reviewers for the thoughtful reviews and helpful suggestions. We have replied to each reviewer's comments individually, and updated the paper with corrections, additional experiments and more explanations based on the reviewer's feedback.